# Adaptive and Neutral Polymorphisms of the *Onne-DAB* Gene from the Major Histocompatibility Complex (MHC) in Sockeye Salmon Populations on the Asian Range

**Anastasia M. Khrustaleva**

Institute of Gene Biology, Russian Academy of Sciences, Vavilova Str., 34/5, 119334 Moscow, Russia; nastia.khrust@gmail.com

**Abstract:** The variability of an MHC complex gene in sockeye salmon populations throughout the Asian range was studied to identify "footprints" indicative of pathogen-mediated selection and neutral demographic processes that have influenced these populations in both the recent and distant past. Genotype frequencies of a haplotype block consisting of two SNP loci (*One_MHC2_109* and *One_MHC2_190v2*) in the *Onne-DAB* gene encoding the β-chain of the MHC class II molecule as well as allelic frequencies of 29 putative neutral SNPs have been traced in 27 sockeye salmon populations in the Asian Pacific coast. Differently directed clines of genetic diversity at the MHC2 loci were observed in sockeye salmon populations inhabiting the Sea of Okhotsk and Pacific coasts of the Kamchatka Peninsula. The formation of these clines can be attributed to a combination of historical processes associated with the colonization of the Asian range and the latitudinal gradient of abiotic and biotic factors influencing the variability of the *Onne-DAB* gene. In continental populations of sockeye salmon, balancing selection was not intense enough to conceal the impacts of demographic and historical processes associated with the fragmentation of the area in the late Pleistocene. In contrast, in island populations, balancing selection effectively maintained the diversity of the *Onne-DAB* gene despite a significant decrease in polymorphism observed in neutral regions of the genome.

**Keywords:** *Oncorhynchus nerka*; sockeye salmon; MHC; SNP; selection; post-glacial colonization

## 1. Introduction

Sockeye salmon (*Oncorhynchus nerka*) is one of the most commercially important species of the North Pacific Rim. Among all the Pacific salmon, it is characterized by the most complex population structure. Isolated populations of different rivers, or metapopulations [1], are often subdivided into seasonal races, which, in turn, turn into local subpopulations and/or groups of subpopulations from different parts of the basins [2,3]. The structure branches up to small panmictic groups reproducing on isolated spawning grounds. Apart from temporal and geographic intraspecific units, different ecological groups are distinguished within sockeye salmon populations—ecological forms (or life history ecotypes) [3–5] and ecotypes reproducing in distinct biotopes [6].

The study of adaptive genetic polymorphism and its role in the evolution and establishment of the population structure of a species is a fundamental problem in population and ecological genetics. Genes of the major histocompatibility complex (MHC) are the most extensively studied adaptive molecular markers in fish [7–9]. MHC is a gene family in vertebrates that plays a key role in the regulation of the immune response. MHC governs the genetic control of interactions among all immunocompetent cells within an organism, enabling the recognition of self and foreign cells and initiating the immune response. This critical mechanism ensures the survival of the species under antagonistic coevolution with a wide range of pathogenic and parasitic organisms. Knowledge of such an important function advances a deeper understanding of evolutionary history, current state, and

prospects for the existence of a population or a species. Considering these substantial traits is of utmost importance when developing fishing strategies and designing sustainable aquaculture practices.

The exons of genes within the MHC complex, specifically those that encode domains containing antigen binding sites known as the peptide-binding region (PBR), play a crucial role in recognizing and binding pathogen proteins on the cell membrane. This recognition subsequently triggers T-helper activation and initiates an immune response. It is commonly observed that these exons exhibit a substantial number of non-synonymous substitutions, high levels of allelic diversity, and heterozygosity [10]. MHC gene polymorphism is maintained through pathogen-induced selection and reproductive strategies for its conservation [10]. Pathogen-dependent mechanisms include both directional selection of individuals that are immune to a specific set of disease agents and balancing selection that supports allelic diversity in the whole population and in subpopulations under conditions of coexistence with a wide range of pathogens [10–13]. Within the framework of the balancing selection hypothesis, two main models are usually considered: overdominance, or the advantage of heterozygotes (the latter have a wider range of presented antigens compared to homozygotes), and an advantage of rare alleles in a parasite-host system, arising as a result of cyclic processes of changes in the parasite's genotype and constant "adjustment" to these changes in the host genotype [10,13]. The mechanisms of sexual selection act mainly through the assortative crossing of individuals differing in their MHC profiles. Experimental data support the preference of female fish for partners that differ as much as possible in MHC gene alleles, which allow offspring to be resistant to a wide range of infectious agents [14,15]. To explain the patterns of variability in the MHC genes, Hedrick proposed the concept of "variation of selection in space and time", according to which their polymorphism is caused by the change of pathogen and parasite communities in different habitats and parts of the range, as well as over the course of historical development [11].

Genes directly involved in immune processes respond quickly to spatial and temporal variations in the diversity of parasites and pathogens, as well as their total load in a particular ecosystem. Thus, it is possible to reveal differences between populations/subpopulations of the host species even with a short divergence time, during which the balance between gene migration and drift has not yet been established, if the gene flow between them is so high that it eliminates differences in neutral loci. In turn, the characteristics of the parasite fauna within specific regions are inherently manifested in the geographical variations observed in allele frequencies of MHC genes. This phenomenon enables the differentiation of distinct local populations and the identification of regional complexes. The utilization of actively expressed genes that exhibit high levels of variability, such as those within the MHC complex, in conjunction with other molecular genetic markers significantly enhances the resolution of genetic population structure analyses for a given species. This enhanced resolution applies to investigations conducted at both local and large-scale geographical levels, facilitating a more comprehensive understanding of population dynamics and evolutionary patterns.

The primary objective of our study was to investigate the variability of a gene within the MHC complex, which plays a crucial role in the immune response in fish, throughout the Asian part of the sockeye salmon range. By examining this gene, our aim was to identify "footprints" indicative of selection and neutral demographic processes that have influenced sockeye salmon populations, both in recent times and in the distant past. Additionally, we sought to assess the significance of balancing selection as a key factor driving the evolution of MHC genotypes and alleles in sockeye salmon. To test the impact of historical and demographic events, we used a set of 29 putatively neutral SNP loci. Therefore, the objectives of this study were to examine the patterns of genetic diversity and differentiation within the Asian range of sockeye salmon, focusing on the *Onne-DAB* MHC class II gene, and to identify the key contributors to the observed patterns of its differentiation.

## 2. Materials and Methods

### 2.1. Study Area and Sample Collection

The samples were collected in 2003 through 2008 in the rivers of the East and West coasts of the Kamchatka peninsula, Chukotka peninsula, mainland coast of the Sea of Okhotsk, Kuril, and Commander Islands (Table 1, Figure 1). Most Kamchatka samples were obtained from fishing companies directly after catches in local fisheries. Sockeye salmon adults were caught using river seine nets in the riverbeds and lake creeks at a distance of 5–30 km from the river mouth during the mass run of sockeye salmon, as well as directly in the spawning lakes (Supplementary Table S1). In the Bolshaya River, juvenile smelt were caught using minnow seine in the upper reach of the Plotnikov River and in the lower course of the Bystraya River (Supplementary Table S1, Figure 1).

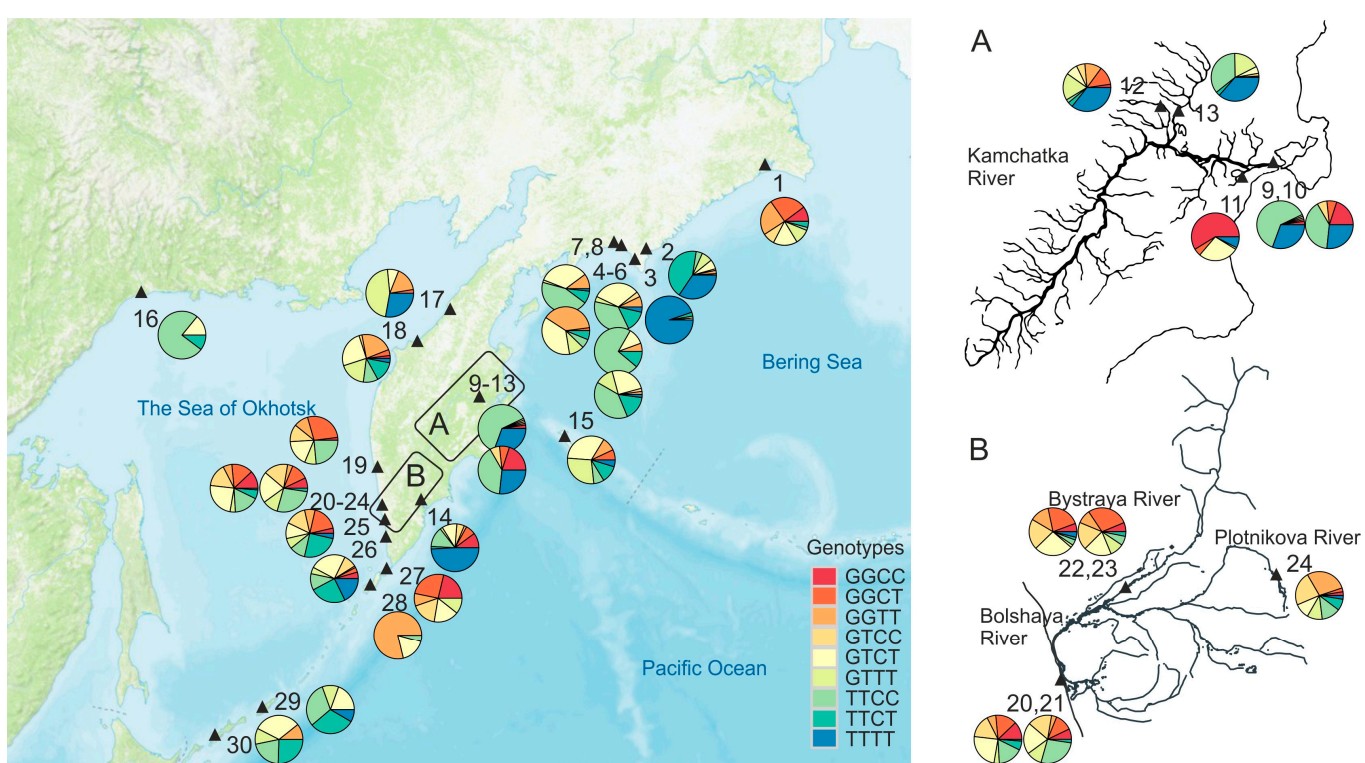

**Figure 1.** Schematic map of the study area with sampling points (▲) and genotype frequencies of the *MHC2* locus in sockeye salmon samples from the Asian coast of the Pacific Ocean. The point's annotations are given in Table 1.

### 2.2. Loci Selection

The variability of two SNP loci—*One_MHC2_190v2* and *One_MHC2_251v2* [16]—located in the *Onne-DAB* gene encoding the β-chain of the class II MHC molecule was analyzed. The first substitution (T/G transversion) is located at position 249 of exon 2, coding the most variable N-terminal tail domain of the β1-chain of the peptide-binding region (PBR); the second one (C/T transition) is at a distance of 61 bp from the first one—within the intron (Supplementary Figure S1). Both loci are included in a 45-locus panel of SNP markers widely used for genotyping sockeye salmon populations in Asia and North America [17–21]. Miller and coauthors [22] demonstrated that the impressive nucleotide diversity of the *Onne-DAB* gene peptide-binding region sequences in the sockeye salmon populations throughout their range is satisfactorily described by the frequencies of five haplotypic lineages. Their amino acid sequences differ by ten substitutions; six of them turned out to be parsimony-informing, while the others were of minor importance. The 6th (substitution E/K(N)) and 10th (substitution Y/D) sites were the most informative. These variable sites are encoded by polymorphic single nucleotide substitutions in the first position of

the codon: *One_MHC2_109* (non-synonymous A/G transition) and *One_MHC2_190v2* (non-synonymous T/G transversion), respectively. From a personal communication of Dr. J.E. Seeb, as well as based on our own data [23] and analysis of sequences deposited in GenBank by Miller et al. [22] (U34707-U34713), it follows that the *One_MHC2_109* and the intronic *One_MHC2_251v2* are linked in the vast majority of the studied populations of sockeye salmon from the Asian and American Pacific coasts. This fact allows us to identify three (1, 2, and 4) of the five lineages (the resolution of the method does not allow us to differentiate lines 3 and 5 from each other while allowing them to be unambiguously discriminated from the others) with more or less certainty by genotyping only two SNP loci, *One_MHC2_190v2* and *One_MHC2_251v2*. Line 1 corresponds to the TC haplotype, line 2—GC, line 4—TT, and lines 3 and 5—GT. Thus, only two well-chosen loci make it possible to more or less confidently characterize the diversity of haplotypic lineages of the *Onne-DAB* gene. Since *One_MHC2_190v2* and *One_MHC2_251v2* are located in close proximity to each other, it is obvious that, unless the opposite is proven, both should be treated as a single linkage group (haplotype block) or a single locus—*MHC2*. Hereinafter, the *One_* prefix is omitted for brevity.

*2.3. Sample Selection*

In our previous papers, in the case of the two rivers (Ozernaya River and Kamchatka River), the most significant reproductive watersheds of the Asian sockeye salmon in Kamchatka, the intra-watershed variability at *MHC2_190v2* and *MHC2_251v2* loci was examined [24,25]. According to our results, it can be postulated that in lake-river systems varying in length and branching, as well as in the number of large lakes, the frequencies of alleles and combined haplotypes of the studied loci vary depending on the river basin peculiarities. Therefore, while in the Kuril Lake basin (Ozernaya River), the samples from different littoral and river spawning grounds did not differ, in the Kamchatka River basin, significant differentiation was observed due to various directional selections in the samples from tributaries of its lower, middle, and upper courses. Thus, for the purposes of our study, it would be quite appropriate in the first case to use a mixed sample from the river mouth. In the second case, our logical reasoning was different. Despite the fact that the frequencies of the *MHC2* locus haplotypes vary so significantly in different localities of this lake-river system, about 80% of the total number of this species in the Kamchatka River reproduces in Azabachye Lake and in middle-course tributaries [26]. Therefore, we confine ourselves to only three samples: from Azabachye Lake and from the two largest middle-course tributaries, the Elovka River and the Dvu 'Yurtochnaya River (see Table 1).

*2.4. SNP Genotyping and Data Borrowing*

Genotyping of *MHC2_190v2* and *MHC2_251v2* SNP loci was conducted using TaqMan-PCR analysis and allele-specific PCR. The genotyping methods are described in detail in [23,27]. The reproducibility of SNP data between the two laboratories was assessed by comparing the individual *MHC2_190v2* and *MHC2_251v2* loci genotypes in two samples, Okh ($n$ = 28) and NKP ($n$ = 43), for which there were overlapping data obtained by both methods. The percentage of discrepancies in the determination of allelic variants varied from 0 to 7% (Supplementary Figure S2), and the average for the *MHC2_190v2* and *MHC2_251v2* loci was 1.9% and 5.7%, respectively. In all cases, genotyping errors were associated with incorrect interpretation of heterozygotes—in 60% of cases, homozygotes determined by the TaqMan approach were identified as heterozygotes using allele-specific PCR, and in 40% of cases, vice versa. Although this level of genotyping error is quite acceptable, if there was duplicated data from the TaqMan-PCR approach, it was used for analysis as it was more reliable.

In this study, we present original data and analyze open data ("http://www.tandfonline.com/doi/suppl/10.1577/T09-149.1?scroll=top (accessed on 10 July 2023)") of Dr. C. Habicht and coauthors [21], based on the genotypic frequencies of the joint locus

*MHC2_190v2−MHC2_251v2* in samples of sockeye salmon from East and North-West Kamchatka. All datasets were combined into a united database of *MHC2* locus genotype frequencies of the sockeye salmon of the Asian coast. In total, 30 sockeye salmon samples from 19 localities (27 populations) on the Asian coast of the Pacific Ocean were analyzed. In addition, we used our data on allele frequencies of putative neutral SNPs (29 loci) (Supplementary Table S2) to trace genetic diversity trends across the Asian range and patterns of population differentiation. In order to select a set of neutral loci, a panel of 45 SNP markers genotyped using the TaqMan-PCR approach [28], widely used for genotyping sockeye salmon populations in Asia and North America [21], was iteratively screened to identify outlier loci at different spatial scales, either by analyzing all population samples together or evaluating various combinations of samples from different locations (Supplementary Figure S3). Outlier detection was performed using coalescent simulations under the hierarchical island model to obtain *p*-values of the locus-specific F-statistic conditioned on observed levels of heterozygosity implemented in Arlequin 3.5 [29].

### 2.5. Statistical Analysis

The estimations of the expected and observed heterozygosities, the inbreeding coefficients ($F_{IS}$), the exact tests on Hardy-Weinberg equilibrium (HWE) using complete enumeration of alleles, the genic and genotypic differentiation (using the exact G-test), and the test on linkage disequilibrium (LD) (using Markov chains) were conducted in GENEPOP 3.4 [30] (two loci were treated separately). The significance level for multiple testing was corrected using the Benjamini-Hochberg FDR method [31]. For 29 neutral loci, the expected heterozygosity (in the sense of Nei's gene diversity) was calculated according to [32], and allelic richness was assessed in FSTAT [33]. Since the gametic phase of the *MHC2* locus alleles was unknown, we used the Expectation Maximization (EM) algorithm [34] for maximum-likelihood haplotype frequency estimation and haplotype imputation implemented in the haplo.stats R library [35]. However, because the EM algorithm assumes Hardy Weinberg equilibrium in haplotype distribution and there is no guarantee that the resulting haplotype frequencies are maximum likelihood estimates, we have chosen to rely on observed genotypic frequencies, and for the further calculations, data on combined genotypes were used. The number of expected multilocus genotypes at the smallest sample size (genotypic richness) based on rarefaction was estimated using the poppr R package [36]. Paired $F_{ST}$ estimations and the Ewens-Watterson neutrality tests were calculated in Arlequin 3.5 [29]. The Mantel-tests for *MHC2* loci and 29 neutral loci were conducted using the ade4 R package [37]. The Cavalli-Sforza chord distances for *MHC2* loci and 29 neutral loci were calculated in the Rphylip package [38]. The graphical projection of the matrices of chord genetic distances based on multidimensional scaling (MDS) was implemented using the MASS library [39] in R. Pearson correlation tests and regression analysis were performed using the ggpubr R package ("https://CRAN.R-project.org/package=ggpubr (accessed on 10 July 2023)") and ggplot2-based data visualization [40]. Possible recent bottleneck events in the populations were tested at 29 neutral loci in Bottleneck 1.2.02 [41] using the three tests: a "sign test", a "standardized differences test", and a "Wilcoxon sign-rank test" assuming the Infinite Allele Mutation model (IAM) for SNP. A bottleneck was considered verified if all three tests were significant. In order to confirm a bottleneck event in the Anana Lagoon population, Garza and Williamson's M-ratio method [42] was applied to the data on the frequencies of 11 microsatellite loci of sockeye salmon for the same sample taken from [43]. The approach calculates M, the mean ratio of the number of alleles to the range in allele size, which can be used to detect reductions in effective population size. In this case, for M-ratios for populations known to have undergone a significant decline in effective population size, the approximate value of M is ≤0.68 [42].

**Table 1.** Sample characteristics, regions and locations, population IDs, and summary statistics for *MHC2_190v2* and *MHC2_251v2* loci and combined genotypes, including the number of expected multilocus genotypes—genotypic richness (*eMLG*), mean expected eterozygosity (*He*), mean observed heterozygosity (*Ho*), the inbreeding coefficients ($F_{IS}$), the results of the linkage disequilibrium tests (LD), the exact tests on Hardy-Weinberg equilibrium (HWE), and the Ewens-Watterson neutrality tests (EW), where a deficiency of homozygotes (*Pr* < 0.05) was considered indicative of balancing selection, while an excess (*Pr* > 0.95) indicated directional selection (in bold). In other test results, an asterisk (*) means that the *p*-value is less than the significance level for accepting the null hypothesis.

| # | Region | Location | Pop. ID | Description | Date of Catch | Source | *n* | *Ho(SD)* | *He(SD)* | *eMLG (SD)* | LD *p*-Value | HWE *p*-Value | $F_{IS}$ | EW *Pr* ($F_{sim} \leq F_{obs}$) |
|---|---|---|---|---|---|---|---|---|---|---|---|---|---|---|
| 1 | Chukotka, Navarin region | Vaamochka Lake | Ch | Adult fish, tributary ecotype | 28 July 2004 | our data | 50 | 0.398(0.049) | 0.049(0.429) | 6.27 (0.88) | 0.6296 | 0.7676 | 0.073 | 0.043 * |
| 2 | | Severnaya Lagoon | KSL | No data | 26 June 2002 | [29] | 98 | 0.328(0.034) | 0.034(0.31) | 4.52 (1) | 0.8598 | 0.0354 | −0.058 | 0.277 |
| 3 | | Anana Lagoon | KAL | Adult fish, early and late run mixed | 24 June 2002 | [29] | 80 | 0.033(0.014) | 0.014(0.033) | 1.75 (0.73) | 0.0771 | 1 | −0.011 | **0.992 *** |
| 4 | Kamchatka peninsula, Olyutor region | Apuka River, Lake Vatit | KAvat | Adult fish, lake form | 7 August 2002 | [29] | 51 | 0.406(0.05) | 0.05(0.41) | 4.81 (0.83) | 0 * | 0.561 | 0.008 | 0.104 |
| 5 | | Apuka River, early run | KAerl | Adult fish, lake form | 24 June 2008–25 June 2008 | our data | 18 | 0.429(0.066) | 0.066(0.438) | 5.37 (0.9) | 0.0006 * | 1 | 0.023 | 0.052 |
| 6 | | Apuka River, late run | KAlt | Adult fish, river form | 24 June 2008–25 June 2008 | our data | 28 | 0.167(0.062) | 0.062(0.244) | 3.79 (0.41) | 0.003 * | 0.245 | 0.325 | 0.774 |
| 7 | | Pakhacha River | KPh | Adult fish, mixed sample | 17 June 2005–27 June 2005 | our data | 59 | 0.39(0.045) | 0.045(0.432) | 4.13 (0.79) | 0 * | 0.5232 | 0.098 | 0.045 * |
| 8 | | Pakhacha River, Lake Potat | KPhpot | Adult fish, lake form | 29 July 2001 | [29] | 50 | 0.459(0.05) | 0.05(0.438) | 4.51 (0.83) | 0 * | 0.9588 | −0.049 | 0.030 * |
| 9 | | Kamchatka River, late run | KK-04 | Adult fish, mixed sample | 29 June 2004–09 July 2004 | our data | 82 | 0.029(0.013) | 0.013(0.273) | 3.03 (0.87) | 0.0046 * | 0 * | 0.893 | 0.367 |
| 10 | | Kamchatka River, early run | KK-05 | Adult fish, mixed sample | 14 June 2005 | our data | 15 | 0.067(0.046) | 0.046(0.434) | 5 (0) | 0.2444 | 0.0001 * | 0.851 | 0.122 |
| 11 | Kamchatka peninsula, East Coast | Azabachje Lake, Bushuyka River | KKa | Spawners, tributary ecotype | 3 July 2004–13 July 2004 | our data | 81 | 0.306(0.036) | 0.036(0.361) | 3.48 (0.75) | 0 * | 0.1817 | 0.152 | 0.407 |
| 12 | | Dvu ′Yurtochnaya River | KKd | No data | 1994, 1995 | [29] | 88 | 0.282(0.034) | 0.034(0.421) | 6.16 (1.03) | 0.0061 * | 0.0002 * | 0.332 | 0.073 |
| 13 | | Elovka River | KKe | No data | 1994, 1995 | [29] | 109 | 0.164(0.025) | 0.025(0.358) | 4.26 (0.88) | 0.0001 * | 0 * | 0.545 | 0.126 |
| 14 | | Avacha River | KAv | Adult fish, mixed sample | 2002 | [29] | 60 | 0.178(0.035) | 0.035(0.437) | 5.66 (1.04) | 0 * | 0 * | 0.595 | 0.131 |
| 15 | Commander Islands | Bering Island, Sarannoye Lake | BS | Adult fish, lake form | July 2008 | our data | 43 | 0.558(0.054) | 0.054(0.474) | 5.78 (0.85) | 0.034 | 0.3525 | −0.18 | 0.015 * |
| 16 | Continental coast of the Sea of Okhotsk | Okhota River | Okh | Adult fish, mixed sample | 22 July 2004 | our data | 80 | 0.192(0.032) | 0.032(0.174) | 2.76 (0.45) | 0 * | 0.9017 | −0.108 | 0.473 |
| 17 | Kamchatka peninsula, North-West | Palana River | KP | Adult fish, mixed sample | 10 July 2003–21 July 2003 | our data | 79 | 0.316(0.037) | 0.037(0.298) | 4.02 (0.67) | 0.1197 | 0.9305 | −0.062 | 0.124 |
| 18 | | Tigil River | KT | No data | 18 June 2002 | [29] | 107 | 0.44(0.035) | 0.035(0.481) | 6.1 (0.99) | 0 * | 0.4659 | 0.086 | 0.002 * |
| 19 | | Bolshaya Vorovskaya River | KV | Adult fish, mixed sample | 17 July 2007–27 July 2007 | our data | 45 | 0.412(0.049) | 0.049(0.489) | 5.75 (0.76) | 0 * | 0.2587 | 0.159 | 0.003 * |
| 20 | | Bolshaya River | KB-03 | Adult fish, mixed sample | 23 July 2003–30 July 2003 | our data | 91 | 0.44(0.039) | 0.039(0.473) | 6.61 (0.95) | 0.0919 | 0.6246 | 0.07 | 0.006 * |
| 21 | | Bolshaya River | KB-04 | Adult fish, mixed sample | 11 August 2004–20 August 004 | our data | 90 | 0.418(0.038) | 0.038(0.468) | 6.11 (0.9) | 0 * | 0.2576 | 0.109 | 0.003 * |
| 22 | Kamchatka peninsula, West coast | Bistraya River | KBb-04 | Smolts, fingerlings | 20 July 2004–12 August 2004 | our data | 33 | 0.484(0.062) | 0.062(0.474) | 6.62 (0.86) | 0.4177 | 1 | −0.022 | 0.014 * |
| 23 | | Bistraya River | KBb-98 | No data | 16 August 1998 | [29] | 56 | 0.5(0.049) | 0.049(0.476) | 6.07 (1.02) | 0.0193 * | 0.9682 | −0.052 | 0.005 * |
| 24 | | Plotnikova River | KBp | Smolts, fingerlings | 9 August 2004–12 August 2004 | our data | 39 | 0.321(0.053) | 0.053(0.504) | 6.59 (0.97) | 0.0047 * | 0.0014 * | 0.367 | 0.000 * |
| 25 | | Opala River | KOp | Adult fish, mixed sample | 17 July 2008–26 August 2008 | ourdata | 31 | 0.429(0.066) | 0.066(0.503) | 7.43 (0.89) | 0.5387 | 0.2473 | 0.151 | 0.001 * |
| 26 | | Ozernaya River | KO | Adult fish, mixed sample | 4 August 2003–07 August 2003 | our data | 95 | 0.45(0.037) | 0.037(0.453) | 6.22 (1) | 0.1058 | 0.8469 | 0.007 | 0.035 * |
| 27 | North Kuril Islands | Shumshu Island, Bettobu Lake | NKS | Adult fish, tributary ecotype | 5 August 2008 | our data | 50 | 0.436(0.051) | 0.051(0.419) | 5.58 (0.56) | 0.8054 | 0.0978 | −0.042 | 0.114 |
| 28 | | Paramushir Island, Glukhoye Lake (Shumnaya Ryver) | NKP | Adult fish, mixed sample | 7 July 2008–13 July 2008 | our data | 48 | 0.17(0.039) | 0.039(0.225) | 2.51 (0.52) | 0 * | 0.0978 | 0.246 | 0.676 |
| 29 | South Kuril Islands | Urup Island, Tokotan Lake | SKU | Adult fish, lake form | July 2008– August 2008 | our data | 35 | 0.5(0.052) | 0.052(0.472) | 4.7 (0.49) | 0 * | 0.7578 | −0.059 | 0.168 |
| 30 | | Iturup Island, Krasivoye Lake | SKI | Adult fish, lake form | 1 October 2006 | our data | 50 | 0.403(0.058) | 0.058(0.382) | 4.72 (0.48) | 0.0102 * | 0.8901 | −0.056 | 0.013 * |

## 3. Results

### 3.1. Tests on Linkage Disequilibrium (LD) and HWE

The *MHC2_190v2* and *MHC2_251v2* loci were polymorphic in all studied populations. It is evident that the *MHC2_190v2* and *MHC2_251v2* loci, situated in close proximity within a single gene, are physically linked. Unexpectedly, the linkage was not supported by tests for linkage disequilibrium in nearly one-third of the samples; a significant association (after FDR-correction) between the analyzed loci was revealed in 19 samples out of the 30 studied in the work (Table 1). One potential explanation for this discrepancy could be the high frequency of point mutations within the sequences encoding the peptide-binding region, coupled with intense positive selection [13,44]. On the other hand, the absence of linkage between two adjacent loci in MHC genes is highly probable, given that the recombination frequency in histocompatibility complex genes is estimated to be significantly higher than the rate of point mutations [45]. This phenomenon is particularly characteristic of exones that are translated into the antigen-binding region of the MHC molecule. Such occurrences primarily arise from micro-recombination events followed by gene conversion [44,46]. Anyway, the spatial distribution of the detected cases of loci linkage in the samples over the range was reticulated and varied from region to region; we can distinguish peculiar local areas of the distribution of nonequilibrium genotypes—the north-eastern coast of the Bering Sea, the Kamchatka River basin, and Southwestern Kamchatka. Noteworthy, in the basin of Kurilskoye Lake (Ozernaya River), the linkage was observed in the tributary samples while the littoral samples were in equilibrium [24].

The concordance of observed and expected genotypic distributions for each locus was revealed in most Hardy-Weinberg equilibrium tests (Supplementary Table S3). Significant deviations from equilibrium, manifested in heterozygotes deficiency, were found only in samples from the mouths of the Avacha River and Kamchatka River at both loci, as well as in samples from the Plotnikova River (basin of the Bolshaya River) and Elovka River (basin of the Kamchatka River) in locus *MHC2_251v2*. For both loci, total deviations from the Hardy-Weinberg equilibrium were observed only in 6 samples out of 30 (Table 1). Deviations from HWE expectations in the form of a deficiency of heterozygotes detected at both loci in samples from the mouth of the Kamchatka River are most likely explained by the Wahlund effect (a heterozygote deficiency in a subdivided population or a mixed sample, where there is variation of allele frequencies between the subpopulations) in mixed catches taken on the migration routes of fish following the separate spawning grounds of a large river basin, where gene drift or differently directed selection can affect these loci. Various forms of selection in the *Onne-DAB* gene in sockeye salmon populations from the Kamchatka River reproduced in tributaries of the upper, middle, and lower reaches were reported in our previous works [24,25].

### 3.2. Genetic Diversity Distribution along the Okhotsk and Pacific Coasts

The smallest intrapopulation diversity (both expected and observed heterozygosity and allelic richness) for two loci and genotypic richness for combined genotypes was characteristic of the Anana Lagoon sockeye population (Table 1, Supplementary Table S2, Figure 2A). Furthermore, extremely low values of observed heterozygosity were noted in samples of early and late sockeye salmon from the mouth of the Kamchatka River, which, among other reasons, can be explained by the Wahlund effect because allelic and genotypic richness in both samples was quite high. The highest estimates of observed heterozygosity were recorded in a sample from Sarannoe Lake (Bering Island), as well as in samples from the Bolshaya River mouth and Iturup Island (Figure 2A). On the contrary, in island populations at neutral loci, there was a tendency to a sufficient decrease in estimates of population diversity.

In addition, opposite trends in population diversity for *MHC2* loci and neutral 29 SNPs are observed for the Kamchatka River basin. While in the *Onne-DAB* gene, intrapopulation diversity was the lowest, the highest diversity in neutral loci was characteristic of the

Kamchatka River populations (Figure 2A). In general, in the Asian part of the range, there was a weak negative correlation between estimates of intrapopulation diversity at the *MHC2* locus ($R = -0.31$, $p = 0.11$ for *Ho* and $R = -0.27$, $p = 0.15$ for genotypic richness) and the geographic latitude of the river mouth, i.e., diversity increased from north to south. Moreover, along the continental Okhotsk Sea coast and West Coast of Kamchatka (West Coast), this trend coincided with the general one, whereas along the Pacific coast (East Coast of Kamchatka and Chukotka), the reverse gradient was observed in the latitudinal direction, i.e., estimates of the heterozygosity for both loci and genotypic diversity of the combined locus decreased to the north (KAL sample as an outlier and island populations due to the special role of neutral demographic processes in them were excluded from the correlation analysis) (Figure 2B).

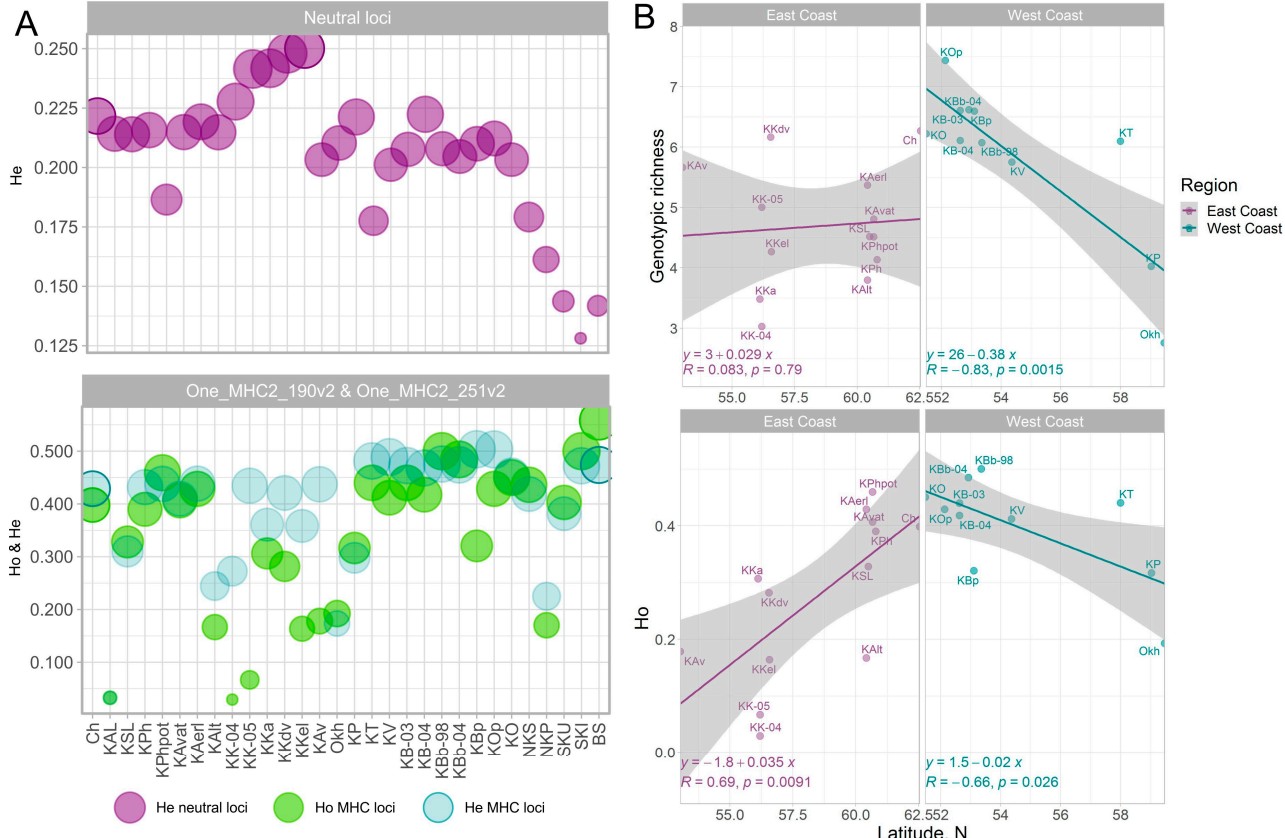

**Figure 2.** Expected heterozygosity for neutral loci and observed and expected heterozygosity for *MHC2_190v2* and *MHC2_251v2* loci in sockeye salmon populations along the Asian Coast of the Pacific Ocean (**A**). Genotypic richness for the *MHC2* locus and observed heterozygosity of the *MHC2_190v2* and *MHC2_251v2* loci as a function of latitude for sockeye salmon from the Asian coast of the Pacific Ocean: East Coast—East Coast of Kamchatka and Chukotka; West Coast—West coast of Kamchatka and the Okhota River. Island populations and sample KAL were excluded from the analysis (see explanation in the text) (**B**).

### 3.3. Population Differentiation

The vast majority of tests for allelic (for both loci) and genotypic (for combined genotypes) differentiation revealed differences between samples from various river basins (after the FDR correction $p < 0.0075$), the only exceptions were samples from geographically close rivers Pakhacha and Apuka (among the samples from lake and river stocks of these watercourses the only sample from the Potat-Gytkhyn Lake located in the upper Pakhacha River was significantly different from the others), the rivers of Southwest Kamchatka (the Bolshaya Vorovskaya River, the Bolshaya River, the Opala River), as well as two

spawning lakes of the South Kuril Islands (Tokotan Lake, Urup Island, and Krasivoe Lake, Iturup Island). In cases of samples collected in the same locality but in different years, no differences were found between them, as well as no significant differences between the samples of juvenile and adult fish caught in the Bystraya River.

The location of the samples in the space of two principal coordinates in accordance with the Cavalli-Sforza chord distances (MDS) calculated by *MHC2_190v2* and *MHC2_251v2* allelic frequencies indicates the proximity of the populations of the West coast of Kamchatka (only one sample from the Palana River was distanced) and a high degree of differentiation of the samples collected at different lake and river spawning grounds of the largest watersheds of Eastern Kamchatka used by sockeye salmon for reproduction (Figure 3A). The Anana Lagoon sample was the most differentiated in the MSD plot by the *MHC2_190v2* and *MHC2_251v2* locus frequencies. As can be seen from the plot on the neutral loci data, all mainland populations form a dense cluster, while the island populations (with the exception of the Shumshu Island sample) are distant from it (Figure 3B).

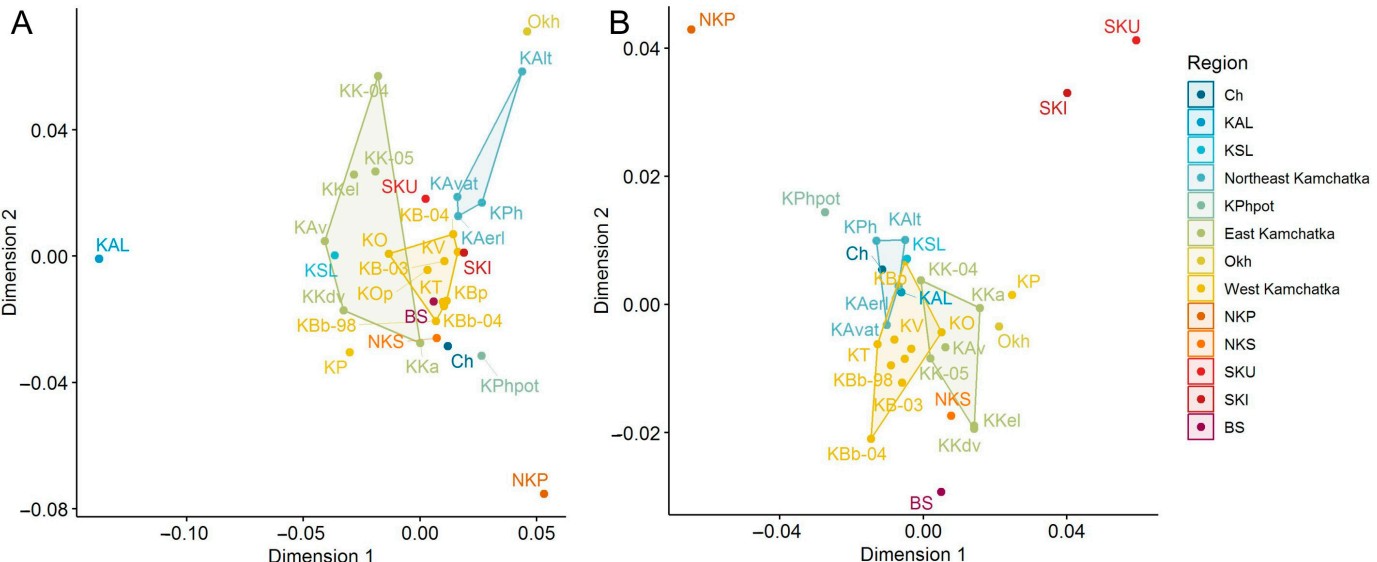

**Figure 3.** Multidimensional scaling plots based on the chord genetic distances for sockeye salmon from the Asian coast of the Pacific Ocean: (**A**)—build by *MHC2* locus genotypic frequencies; (**B**)—build by 29 neutral loci allelic frequencies. Regions are marked by different colors; some populations are combined into polygons according to region, and some are outliers.

The interpopulation genetic diversity estimated by the $F_{ST}$ value averaged 0.234 ($p = 0$). According to the paired $F_{ST}$ sockeye salmon populations of the rivers Laguna-Anana (Olyutorsky region), Shumnaya (watershed of Lake Glukhoye, Paramushir Island), as well as Palana (Western Kamchatka), Okhota (mainland coast of the Sea of Okhotsk), and Azabachye Lake (Kamchatka River basin), they differed to the greatest degree from all other samples (Figure 4A). The samples from the Southwestern Kamchatka rivers were the closest to each other, as were samples from other lake-river systems of the West Kamchatka coast (Tigil River) and the nearest island of the Kuril Ridge—Shumshu.

The significance of the correlation between the matrices of genetic and geographical distances both throughout the Asian part of the range and separately along the coasts of the Sea of Okhotsk (West Coast) and the Pacific Ocean (East Coast) was assessed using the Mantel-test (all Asian coasts $p = 0.338$; West Coast (including Kuril Islands) $p = 0.061$; East Coast (including Kuril Islands) $p = 0.864$; West Coast (excluding Kuril Islands) $p = 0.002$ **; East Coast (excluding Kuril Islands) $p = 0.511$, where ** means $p < 0.01$). The test results indicate a mismatch between the population structure of the Asian sockeye salmon revealed by the variability of the *MHC2* locus and the isolation by

distance model (with one exception—for the West coast of Kamchatka), i.e., the absence of a clear correlation between the magnitude of gene migration and the proximity of the tributaries to each other. Nevertheless, by neutral SNP, all the correlations along both coasts were significant (all Asian coasts $p = 0.2$; West Coast (including Kuril Islands) $p = 0.0182$ *; East Coast (including Kuril Islands) $p = 0.0003$ ***; West Coast (excluding Kuril Islands) $p = 0.0062$ **; East Coast (excluding Kuril Islands) $p = 0.0002$ ***, where * means $p < 0.05$, ** — $p < 0.01$, *** — $p < 0.001$).

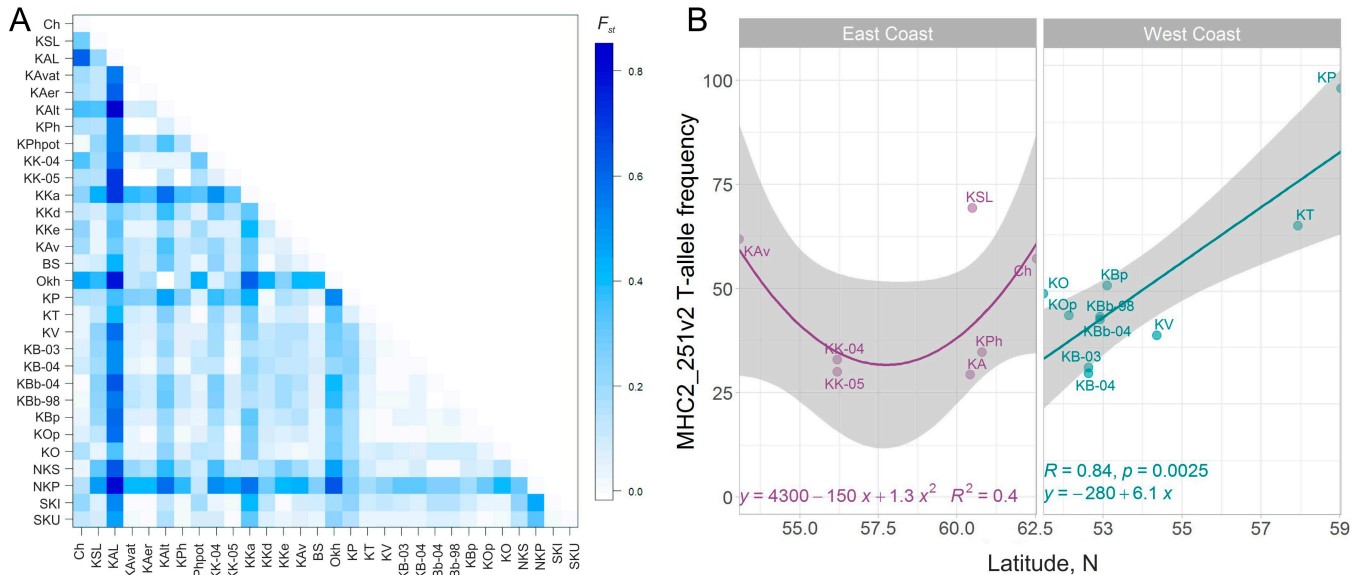

**Figure 4.** (**A**)—Pairwise $F_{ST}$ matrix for sockeye salmon samples from the Asian coast of the Pacific Ocean. Rows/columns of the heatmap are sorted by the latitude/coast location of the watershed. (**B**)—*MHC2_251v2* T-allele frequencies as a function of latitude for sockeye salmon from the East and West Kamchatka Coasts—East Coast and West Coast, respectively.

### 3.4. Clines of Alleles, Genotype Frequencies, and Neutrality Tests

In general, the frequency distribution of genotypes of the combined *MHC2* locus over the Asian part of the range is rather mosaic (Figure 1, Supplementary Figure S4). Nevertheless, along the West coast of the Kamchatka peninsula and partly along the East Coast of Kamchatka (if we analyze only the samples from the estuaries of large rivers and exclude the KAL sample as an outlier), trends in the allelic frequency distribution of the *MHC2_251v2* intron locus are observed: the frequency of the T-allele significantly increases in the direction from south to north (Figure 4B). For the East Coast, this trend is not significant (for a linear relationship, $R^2 = 0.002$, $p = 0.93$). However, if we approximate the dependence of the T-allele frequency on the latitude of the river mouth with a quadratic polynomial function, then this dependence becomes significant ($R^2 = 0.403$, $p = 0.036$) (Figure 4B). Furthermore, the concavity of the relationship seems to be driven by a single population, and therefore, it needs to be clarified, and the number of samples should be increased for a more comprehensive analysis. It can also be noted that genotype distributions are more homogeneous along the Sea of Okhotsk coast than along the Pacific coast, where the frequencies of the TTTT-genotype are most unevenly scattered.

According to the Ewens-Watterson neutrality test, in 13 out of 30 localities, the effect of balancing selection at the *MHC2* locus can be traced (Table 1). In some of the lake-river systems, the genotype frequency distributions appeared to be in agreement with the expected ones according to the infinite allele model, i.e., the analyzed locus behaved neutrally. Only one sample from the Anana Lagoon, where a decrease in genetic diversity and high differentiation were previously revealed, was characterized by a sharply asymmetric distribution of genotype frequencies (in the Ewens-Watterson tests)

and a high probability of dominance of the most adapted genotype, which is typical for loci evolving under directional selection.

*3.5. Detecting Population Decline with Bottleneck Tests on Neutral Markers*

The results of the bottleneck tests for 29 putative neutral loci are quite inconclusive (Supplementary Table S4). Only for five samples out of thirty, none of the tests showed the presence of a bottleneck, and four of them are rather small, isolated island populations, for which periodic fluctuations in abundance are very probable events. Nearly half of the populations in all three tests showed recent declines, including the Anana Lagoon population. To verify the significant impact of the loss of allelic diversity in the Anana Lagoon population due to bottleneck, we analyzed the literature data on the frequencies of 11 microsatellite loci of sockeye salmon for the same sample [43]. M-ratio scores (Supplementary Table S5) were below the threshold value of 0.68 for 6 out of 11 loci, which by implication may indicate that this population has recently passed the bottleneck. However, the mean value for all loci was 0.695, which is higher than the accepted threshold. Thus, it is impossible to definitively identify the probable bottleneck signature in the Anana Lagoon population from the available microsatellite data.

## 4. Discussion

*4.1. Demographic Factors Potentially Affecting the Clinal Character of Population Diversity Variation along the Asian Coast of the Pacific Ocean*

Distinct clines of genetic diversity at the *MHC2* loci were observed in sockeye salmon populations inhabiting the Okhotsk and Pacific coasts of the Kamchatka Peninsula. Such clinal patterns can be attributed to demographic (effective population size, gene migration, etc.) or historical processes associated with the formation of the range of this species in Asia (postglacial colonization, secondary contact), as well as the latitudinal gradient of various abiotic and biotic factors affecting the variability of the *Onne-DAB* gene. According to the "abundant center-periphery" hypothesis, in the central populations, the highest effective size (Ne) and gene flow (m) are observed, reflecting high values of intrapopulation variability and low interpopulation differentiation [47,48]. Meanwhile, towards the range edge, genetic diversity tends to decrease while population divergence increases. Our findings provide confirmation of this hypothesis, specifically in relation to neutral SNPs. We observed that the indices of intrapopulation diversity for neutral markers were consistently lower in peripheral island populations compared to continental populations. However, there was neither any clear dependence of indices of sample differentiation (paired $F_{ST}$ and genetic distances) nor heterozygosity or allelic/genotypic diversity of the *MHC2* loci on the effective number and distance from two core populations of this species in Asia (the two largest stocks of the Western (Ozernaya River) and Eastern (Kamchatka River) Kamchatka) (see Figure 2).

The ladder model assumes that migration primarily occurs between neighboring populations, particularly in the case of a belt range or one-dimensional model, and given the strong homing of this species, gene flow between populations is expected to decrease as geographical distance increases. Accordingly, distance-based isolation was observed at neutral SNP loci along the West Pacific coast. However, tests conducted using the data from the combined *MHC2* locus did not provide support for isolation by distance patterns, both in the Sea of Okhotsk and on the Pacific coast. These findings suggest that gene migration is not the primary factor influencing the variability of this specific marker within the Asian range of sockeye salmon.

*4.2. Temperature and Pathogens as Factors Maintaining the Latitudinal Gradient of the Onne-DAB Gene Variability*

The clinal distribution of intrapopulation diversity indices observed along the continental coast is likely influenced by factors that exhibit variation in a latitudinal direction, such as temperature and/or the distribution of specific pathogenic agents. Theoretical considerations suggest that polymorphism within MHC genes should generally decrease

from south to north due to a reduction in species diversity and a weakening of pathogen virulence in high-latitude regions. This decrease in virulence can be attributed to factors such as lower metabolic intensity, shorter growing seasons, and reduced rates of ontogenesis and mutagenesis under colder climatic conditions. In essence, temperature contributes to the gradient of pathogen-induced balancing selection pressure on MHC genes, which manifests as a latitudinal pattern.

On the example of Atlantic salmon, the increase in allelic and nucleotide diversity, as well as heterozygosity (cited by [49]), in the β2-exon sequences of the DAB gene encoding the peptide binding region was demonstrated with the rise in water temperature manifesting in the latitudinal gradient of these parameters [50]. However, no correlation between estimates of intrapopulation variability ($n_a$, *He*) of the β1-exon of the same gene and the latitude of the native river basin was found for the sockeye salmon throughout the entire range [51]. In addition, no significant relationship was found between allelic and nucleotide diversity of β1-exon sequences and water temperature in the system of tributaries and lakes of a large lake-river system (Wood River, Alaska) [49]. At the same time, estimates of the expected heterozygosity of the sockeye salmon from Wood River were significantly correlated with temperature at spawning grounds [49]. In addition, significant differences were demonstrated in the variability of the *MHC2* locus on a small geographical scale (within one river basin), comparable to those observed throughout the entire range of this species, according to the authors, due to the high isolation of lake populations and the specific composition and virulence of pathogen and parasite communities in each lake [12,49]. Previous studies conducted in the Kamchatka River drainage have identified high levels of population differentiation for the two examined loci, as well as significant variations in estimates of intrapopulation diversity among samples collected from different tributaries within the river basin [24,25]. Furthermore, a positive correlation between allelic diversity and water temperature at the channel's inflow of the respective tributary has been shown.

Nevertheless, the contrasting trends observed in diversity index distribution among West and East Kamchatka suggest that factors other than just temperature gradients in the latitudinal direction contribute to the observed pattern of genetic differentiation in sockeye salmon along the Asian Pacific coast. Undoubtedly, one of the most influential factors is the temporal and spatial distribution and species composition of parasitic and pathogenic organisms in the study area, since according to generally accepted concepts, the MHC class II genes involved in the formation of immunity to extracellular infections evolve under the influence of pathogen-induced selection [10].

At this stage, it is difficult to draw any conclusions about the role of various infectious agents in the pattern of *Onne-DAB* gene variability. However, it can be assumed that the main impact is exerted by representatives of the freshwater parasite fauna, particularly common lacustrine species such as myxosporidia, *Myxobolus* sp., and cestodes, *Diphyllobothrium* sp. [52]. Their ratios in different lakes vary significantly, and they can be considered indicator parasites of lacustrine stocks of sockeye salmon [53]. For example, in Kurilskoye Lake, juvenile sockeye salmon were characterized by the highest degree of both parasite invasions on the Asian Coast, while in Azabachje Lake, their prevalence was much lower [52]. In addition, there were no cases of IHNV (Infectious Hematopoietic Necrosis Virus) in juveniles in Azabachje Lake, while in Kurilskoye Lake, this pathogen was detected annually in 25% of juveniles and in more than 60% of brood fish. Genotype spectra in the samples from Kurilskoye Lake and Azabachje Lake also differed; the proportion of T-alelle in the *MHC2_251v2* locus was two times higher in Kurilskoye Lake. There are high chances that the notable frequency gradient of the *MHC2_251v2* T-allele observed along the Okhotsk Sea coast can be likely attributed to the clinal distribution of a specific pathogen, the prevalence of which varies with latitude. The association of individual alleles of the MHC complex genes in some salmon species with resistance/vulnerability to specific pathogens has been repeatedly demonstrated in experimental studies [54–58]. Nevertheless, based on the existing data, it is not yet feasible to definitively ascertain the adaptive

significance of a particular allele or genotype, as each of them encompasses multiple cryptic sequences responsible for encoding the peptide-binding region of the MHC class II (MHCII) molecule. In previous studies focused on sockeye salmon, a range of 10 to 27 haplotype (allelic) variants have been previously identified within the sequences that encode the β1 domain of the MHC class II molecule [12,22,51].

*4.3. Different Types of Selection Identified in Asian Populations*

The trends of geographical distribution in the genetic variation across the range and the population differentiation pattern, which are incongruent with those of neutral genetic markers, provide evidence for the influence of selection on MHC loci. According to the results of neutrality tests in almost half of the localities, signs of balancing selection at the *MHC2* locus were revealed. However, the Evens-Watterson tests were not always significant, and in some samples, the *MHC2* locus behaved rather neutrally. Our new and already published data [24,25] are quite consistent with the generally accepted ideas about the predominance of balancing selection over the entire range, in some regions or in large population systems in general, and the action of directional selection, leading to adaptive divergence of populations in some isolated localities: lakes, tributaries of the upper reaches of large river basins, etc. [12,22].

Only one sample from Anana Lagoon Lake demonstrated the pattern of directional selection based on neutrality tests. The shift in genotype frequencies in the sample indicates either a relatively recent epizootic in this lake-river system or the limited species composition of pathogens and parasites in this water body. There is only fragmentary information about the planktonic community of Anana Lagoon Lake [59]. The composition of zooplankton is very specific here: the abundance of *Cyclops scutifer* is low, and the main food organism for juvenile sockeye salmon is the brackish-water cold-tempered species *Limnocalanus macrurus*, which is not an intermediate host for Diphyllobothriidae. Therefore, it can be assumed that in this lake, the infestation of fish with cestode plerocercoids should be low, and other parasitic organisms should come to the fore. According to the data of Bugaev and Kirichenko [60], the density of oligochaetes in benthos samples was quite high, which may indicate the presence of myxosporidia in the lake because their actinosporeic phase takes place in the bodies of Oligochaeta of the family Tubificidae [52]. On the other hand, similar effects in natural populations can be explained by demographic processes associated with a high degree of isolation of the lacustrine sockeye and a decrease in the effective population size (genetic drift, bottleneck). Based on the findings from bottleneck tests conducted on neutral markers within this population, it is difficult to unambiguously identify the determinant effect of a reduction in abundance on allelic diversity. Bottleneck tests were significant in almost half of the studied populations, which, however, did not lead to a significant shift in allelic frequencies in any of them, nor was the directional selection confirmed in any of them. Moreover, the samples from this lake-river system, according to the results of the 13 microsatellite loci polymorphism analysis [61] as well as our neutral SNP data, were not characterized by a high degree of differentiation and were clustered together with samples from neighboring rivers in the Olyutor region. Thus, we cannot expect a significant contribution from genetic drift or a bottleneck in genotypic distribution bias in the Laguna Anana sample. These data give us sufficient evidence to support the concept of directional selection in favor of the most adapted haplotypes in the population.

*4.4. Parallelism in Patterns of Onne-DAB Gene Diversity Distribution and Post-Glacial Colonization on the Asian Part of the Sockeye Salmon Range*

Patterns of postglacial sockeye salmon colonization of the Asian coast during the late Pleistocene may be reflected in a contemporary gradient of intrapopulation variability, which exhibits a correlation with the likely directions of ancient dispersal flows [47]. According to the available data on mtDNA and microsatellite loci variability in sockeye

salmon in the Russian Far East, the following picture emerges: the colonization of the West coast of Kamchatka most likely proceeded sequentially from south to north [61,62], while the colonization of the East Coast followed a more complex scenario. Both types of data suggest that the dispersion of sockeye salmon populations along the West Pacific coast originated from multiple refugees. One of them is believed to be the paleobasin located in the middle sections of the Kamchatka River. Additionally, there is partial evidence indicating a northern origin, likely stemming from the Beringia refugium [61–63]. The tendencies of changes in the intrapopulation diversity indices in both loci and in allele frequencies, at least in the intronic *MHC2_251v2* locus in the study area (with a number of exceptions), are quite consistent with the prevailing ideas about the formation of the range of this species in Asia. Apparently, the effects of global climatic oscillations and associated evolutionary and demographic events, despite the leading role of adaptive transformations in the analyzed genome region, still leave an imprint on the overall pattern of its variability. The balancing selection that maintains diversity in modern populations, which we have identified in some samples, is not strong enough to mask the effects of neutral and historical processes. Some researchers came to similar conclusions when studying the polymorphism of the MHC complex genes in various species of mammals and birds [64].

### *4.5. Island Populations*

Regarding island populations, our analysis of neutral SNPs as well as literature data on neutral genetic markers (microsatellites) [65] indicates a notable reduction in genetic diversity within island sockeye populations compared to continental populations. The observed low levels of intrapopulation polymorphism and significant genetic divergence among the island populations based on neutral markers can be attributed to several factors. These factors include genetic drift occurring during extended periods of isolation, small effective population sizes leading to increased inbreeding, frequent bottleneck events due to the greater amplitude of abiotic factors during global climate fluctuations, and the "founder effect". Contrary to expectations, our data reveal that, in the majority of cases, island populations of sockeye salmon exhibit elevated levels of intrapopulation diversity at the *MHC2* locus. These levels are well comparable to, and in some instances even surpass, the corresponding estimates observed in continental populations. This circumstance indicates the dominant role of stabilizing selection in the formation of the modern diversity of the *Onne-DAB* gene in the island populations of the Asiatic sockeye salmon. Moreover, its effect is so strong that it fully compensates for the loss of diversity due to neutral demographic processes. While in the populations of the Northern Kuril Islands one of the genotypes dominates in number due to gene drift or the predominance of a certain pathogen/community of pathogens in nursing lakes, in the Southern Kuril Islands the genotypes are more evenly distributed. Perhaps this is due to the fact that the sockeye populations of the southern islands live in the contact zone of the Arctic and boreal fauna; accordingly, the species composition of parasites is much wider here, and, therefore, local balancing selection is more intense. Our results do not agree with those obtained earlier for island populations of tuatara, according to which population bottlenecks and isolation have a larger influence on patterns of MHC variation in such populations than selection [66].

### 5. Conclusions

In summary, our findings provide further support to previous conclusions regarding the prevalence of balancing selection driven by pathogen-mediated pressures on the genes of the histocompatibility complex in sockeye salmon populations on a large-scale geographical scale (at the level of regions and river basins) and a variety of forms of selection on a local scale (in isolated lakes and tributaries). Differently directed clines of genetic diversity at the *MHC2* loci revealed along the East and West Kamchatka coasts may be caused by a latitudinal gradient of abiotic and biotic factors that apparently affect

the variability of the *Onne-DAB* gene. In addition, some historical processes associated with the colonization of the Asian range, such as post-glacial expansion and secondary contact events, likely contribute to the observed clines of genetic diversity. While the balancing selection at *MHC2* loci in continental sockeye salmon populations was not sufficiently strong to mask the effects of demographic and historical processes associated with range fragmentation and subsequent restoration during the Late Pleistocene, the island populations exhibited a contrasting scenario. In these populations, the balancing selection effectively maintained the diversity of the *Onne-DAB* gene, despite a significant reduction in polymorphism observed in neutral regions of the genome. All these findings highlight the complex interplay between selection, demographic processes, and historical events in shaping the genetic diversity and differentiation of the *Onne-DAB* gene in sockeye salmon populations in Asia.

**Supplementary Materials:** The following supporting information can be downloaded at: https://www.mdpi.com/article/10.3390/d15070853/s1. Figure S1. Scheme of the *Onne-DAB* gene in sockeye salmon, location of the *One_MHC2_190v2* and *One_MHC2_251v2* loci, and designation of haplotypes and genotypes of the combined *MHC2* locus. Figure S2. Percentage of coincidence of allele scores for the *MHC2_190v2* and *MHC2_251v2* loci in two samples, Okh and NKS. Figure S3. Examples of output images for outlier-SNP detection tests using Arlequin 3.5 for two combinations of samples: (A)—all continental samples vs. all island populations; (B)—all samples vs. BS sample. Loci falling above 5% (in the upper part of the graph) and below 1% (in the lower part of the graph) quantile limits were reclassified as outliers. Here, *serpin*, *HGFA*, and *GPDH* are considered outliers. Figure S4. Plots of genotype frequencies as a function of the river mouth latitude for sockeye salmon from East (A) and West (B) Kamchatka coasts. Table S1: Sampling regions and locations, population IDs, and sampling methodology: date and place of catch, coordinates of a river mouth or a lake of catch, fishing gear, and collector name if known. Table S2: Characteristics of 29 putative neutral SNP loci. $n_a$—mean allelic reachness; *He*—mean expected heterosigocity. Table S3: Population IDs and summary statistics for the two MHC loci, including allelic richness ($n_a$), expected heterozygosity (*He*), observed heterozygosity (*Ho*), the inbreeding coefficients ($F_{IS}$), and the results of the exact tests on Hardy-Weinberg equilibrium (*HWE*). In HWE test results, an asterisk means that the *p*-value is less than the significance level for accepting the null hypothesis. Table S4: Results of bottleneck tests using 29 putative neutral loci: *N*—mean sample size, $n_a$—mean allelic reachness, *He*—mean expected heterosigocity, p_sign_IAM—results of the "sign test" for loci evolving under the Infinite Allele Model (IAM); p_stdv_IAM—results of the "standardized differences test", IAM; p_W_1t_IAM and p_W_2t_IAM—results of one-tail and two-tail "Wilcoxon sign-rank tests", IAM. Table S5: Results of bottleneck tests based on M-ratio for 11 microsatellite loci. *L1, L2*—sizes of the largest and smallest alleles; *l*—repeat length; *k*—number of repeats; *r*—difference between the sizes of the largest and smallest alleles expressed in repeat numbers.

**Funding:** The research was supported by the Russian Science Foundation (project No. 23-24-00307).

**Institutional Review Board Statement:** This study was carried out in compliance with Federal Law No. 498-FZ 'On Responsible Handling of Animals and on Amending Certain Legislative Acts of the Russian Federation' (17 December 2018). No ethical approval was required for dead fish provided by local fisheries.

**Data Availability Statement:** The datasets generated and analyzed during the current study are available from the author on reasonable request.

**Acknowledgments:** Author gratefully acknowledges to J.E. Seeb (College of the Environment, University of Washington, Seattle, WA, USA) for his comprehensive assistance, provision of methods, and laboratory analysis management, as well as to all the staff of the Environmental Genomics Laboratory, Department of Hydrobiology and Fisheries, University of Washington, M.V. Shitova, (The Vavilov Institute of General Genetics of the Russian Academy of Sciences, VIGG RAS) and P.C. Afanasyev, (Russian Federal Research Institute of Fisheries and Oceanography, VNIRO) for their help in laboratory processing, A.A. Volkov (VNIRO) for the allele-specific PCR method adapting, primers design, and technical support, N.V. Klovach (VNIRO) for sample collection management, and the employees of VNIRO, Kamchatka branch of the VNIRO (KamchatNIRO),

Pacific branch of the VNIRO (TINRO), Sakhalin branch of the VNIRO (SakhNIRO), and Magadan branch of the VNIRO (MagadanNIRO), who took part in the samples collection.

**Conflicts of Interest:** The author declares no conflict of interest.

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
