# Peer review of "Adaptive and Neutral Polymorphisms of the Onne-DAB Gene from the Major Histocompatibility Complex (MHC) in Sockeye Salmon Populations on the Asian Range"

_diversity, doi:10.3390/d15070853_

Round 1

Reviewer 1 Report

Major comment

1. Overall, the author may have neutral SNP data, but not used them for population genetic analysis and discuss only MHC2 loci. Very confusing.

2. Discussion is too long and redundant, and it is difficult to follow. Discussion should be much shorter.

Minor comments

1. Figure 1. There are MHC2 haplotypes constructed by four SNPs.  I did not followed why there are four loci.  

2. There are no neutral SNP information: which loci, where they located, and how to genotype.

3. MDS plotting was done using MHC2 loci, and author discuss population structure of sockeye salmon from two loci which are tightly linked.

4. Why did not you use neutral SNP data for MDS in addition to MHC2 loci.

5. Discussion 4.3

  Author discuss pathogen-induced selection in some analyzed population.

  Is there any evidence of disease outbreak?

  In addition, if so, possibly bottleneck will be observed. Better to calculate bottleneck using neutral SNPs. 

6. Discussion 4.4 

  Understood there is directional selection in Anana Lagoon Lake in this study. But this result is opposite to the previous studies using microsatellite. Author should re-confirm population genetic structure and difference using neutral DNA markers of your data.

7. Conclusions

  As same as discussion sections, too long. One paragraph is enough for this section.

Author Response

I'm very grateful to the esteemed reviewer for his/her valuable comments and recommendations. I took into account most of the recommendations and revised the text in conformity with all the points of the review.

Major comment

  1. Overall, the author may have neutral SNP data, but not used them for population genetic analysis and discuss only MHC2 loci. Very confusing.

I agree with the reviewer that a comprehensive study of the population structure of this species using both MHC2 loci and neutral SNP data would provide more information. But I would like to confine myself to the scope of the tasks of this article and focus only on elucidating the diversity of MHC loci in the Asian range, the features of differentiation of sockeye salmon populations according to these loci; identification of key factors influencing the observed patterns of geographical distribution of its genetic diversity. I hope that we will be able to carry out a more detailed population genetic analysis of this species in the future.

  1. Discussion is too long and redundant, and it is difficult to follow. Discussion should be much shorter.

Thank you for recommendation, I have tried to shorten this section.

Minor comments

  1. Figure 1. There are MHC2 haplotypes constructed by four SNPs.  I did not followed why there are four loci.  

I corrected the Figure 1 and added a scheme of the Onne-DAB gene with designation of haplotypes and genotypes of the combined MHC2 locus in Supplementary file.

  1. There are no neutral SNP information: which loci, where they located, and how to genotype.

Thank you for this recommendation. I have added a table with the characteristics of the 29 putative neutral SNP loci in the Supplementary. Some information about their genotypic method and selection method has been added to Materials and Methods, lines 186-196.

  1. MDS plotting was done using MHC2 loci, and author discuss population structure of sockeye salmon from two loci which are tightly linked.

I'm sorry that the Figure capture was wrong. The Figure 3B is the MDS plot build from chord distances calculated on the base of 29 neutral loci allelic frequencies.

  1. Why did not you use neutral SNP data for MDS in addition to MHC2 loci.

It's undoubtedly important and I compared population differentiation pattern obtained with two types of markers in order to provide evidence for the influence of selection on MHC loci.

  1. Discussion 4.3

  Author discuss pathogen-induced selection in some analyzed population.

  Is there any evidence of disease outbreak?

  In addition, if so, possibly bottleneck will be observed. Better to calculate bottleneck using neutral SNPs. 

Unfortunately, we do not conduct systematic observations of the epidemic situation in the watersheads of sockeye salmon reproduction. There is only fragmentary information about the infection of juveniles and spawners with some types of pathogens in several years (Sergeenko et al., 2013).

  1. Discussion 4.4 

  Understood there is directional selection in Anana Lagoon Lake in this study. But this result is opposite to the previous studies using microsatellite. Author should re-confirm population genetic structure and difference using neutral DNA markers of your data.

According to this and previous recommendations I added an analysis of possible bottlenecks using neutral SNPs and literature data on 11 microsatellite loci in Anana population. The relevant tables are placed in Supplementary, added special sections to Materials and Methods and Results.

  1. Conclusions

  As same as discussion sections, too long. One paragraph is enough for this section.

Thank you for the comment. The text was reduced.

I appreciate the special attention to the manuscript and the time spent on such valuable comments.

With my great respect and gratitude,

Anastasia Khrustaleva

Reviewer 2 Report

Author Khrustaleva provides a comprehensive population genetic analysis of sockeye salmon genotype data for 2 MH gene SNPs (with 29 neutral SNPs for comparison) across their Asian range (27 populations). The dataset is comprised of original (new) data as well as previously published data.  The compilation of these data is not only impressive, but also important for both conservation/management applications, and for our understanding of the evolution of Asian sockeye salmon. The paper is generally well written, although some language issues make it hard to follow in places.  The paper does need to be better organized, and the Discussion is perhaps too long. Overall, this is an important paper and will be of interest to a diverse readership.  Below I give more specific comments.

General

1) In the Introduction, the author describes the diverse arrays of ecotypes and life histories observed in sockeye salmon; however, the author does not specifically test for patterns of MH (or neutral) SNP genotype divergences among ecotypes or life histories. I suggest either reducing this Introduction text, or adding ecotype/life history as factors in the analysis of observed variation in MH genotypes.

 2) Detailed justification for how specific river systems were samples is provided (based on a previous publication); however, the text appears to apply to a total of 4 of the 29 samples. I suggest adding a supplementary file giving justification for the sampling methodologies for all included “populations” – this is important information for the interpretation of the population genetic analysis results.  

 3) A technical point: samples were collected over multiple years and genotyping was performed in more than one lab. While SNP genotyping is generally robust, I suggest adding an analysis of temporal and lab effects of genotype frequencies, just to eliminate the question of possible bias.

 4) This study includes two types of SNP data: MH genotypes and putative neutral genotypes. I found it hard to follow in the Results which type (or both) of data were being reported.  I suggest adding text to always make it clear which SNPs are being referred to.

 5) The first section of the Discussion deals with the unexpected low LD between the two MH SNPs in many populations. While this is curious, it was not the aim of this work and it certainly is not the main outcome.  I suggest reducing this Discussion and moving it lower in the Discussion.

 6) I like the Discussion of the observed clines in MH diversity, and both temperature and pathogen community could contribute to that pattern.  However, the author does not present actual pathogen data, hence I suggest this discussion be a bit more cautious (although the author does acknowledge that other factors may play a role). In any case, the author should consider putting this discussion text close to the beginning of the Discussion.

 7) The first line of the Conclusion indicates that these data support balancing selection as a key factor in MH genotype/allele evolution in sockeye salmon. I suggest this be added as a key goal of the paper in the Introduction. As it stands the paper is framed as a report of observations of genotype/allele frequency distributions; however, the analysis is much more than that!

 8) While the paper is generally well-written, I suggest the grammar and word use be carefully checked – in places it is hard to follow and in others I think the text means something different than the author intended.

Specific comments

1) The final sentence of the abstract is hard to follow and needs to be clarified.

2) The author rarefied the data to N >9 based on the smallest samples size, but in Table 1, the smallest reported is 15.  Furthermore, most sample sizes were much larger than that, so it seems overly cautious to rarefy to N=10.

3) This study includes 29 putatively neutral SNPs used as the neutral control data; however, it seems some of those SNPs may not be entirely neutral.  I think text needs to be added describing the neutral SNPs and an assessment of whether they are evolving by drift only added.

4) Figure 4B: the concave nature of the relationship between allele frequency and latitude seems to be driven by a single population, perhaps the author should be cautious in interpreting this pattern.

5) Line 406-407: I don’t recall seeing the results of the Mantel test for the neutral SNP data?  It may have missed it as I found it hard to know which data type was being reported.

6) Figure 4: perhaps a typo?  “hitmap” = heatmap

7) Line 503: I suggest sticking with the generally accepted significance report (not P>0.99)

8) Lines 535-536: “colonization by small number and less polymorphic…” why would the colonizing individual be less polymorphic? Generally the range edge individuals are more diverse, at least at MHC loci.

As outlined above, the English in this paper is strong, but does need considerable editing to improve clarity.

Author Response

The author is very grateful to the esteemed reviewer for his/her special attention to the manuscript and the detailed analysis of the work. I absolutely agree with all the comments made and consider them extremely valuable for improving the text of the manuscript. Many comments made me think and revise some of points, which will undoubtedly improve the presentation of this work.

1) In the Introduction, the author describes the diverse arrays of ecotypes and life histories observed in sockeye salmon; however, the author does not specifically test for patterns of MH (or neutral) SNP genotype divergences among ecotypes or life histories. I suggest either reducing this Introduction text, or adding ecotype/life history as factors in the analysis of observed variation in MH genotypes.

Thank you for the comment. The text was reduced.

 2) Detailed justification for how specific river systems were samples is provided (based on a previous publication); however, the text appears to apply to a total of 4 of the 29 samples. I suggest adding a supplementary file giving justification for the sampling methodologies for all included “populations” – this is important information for the interpretation of the population genetic analysis results. 

It's undoubtedly important, the table was added to Supplementary.

 3) A technical point: samples were collected over multiple years and genotyping was performed in more than one lab. While SNP genotyping is generally robust, I suggest adding an analysis of temporal and lab effects of genotype frequencies, just to eliminate the question of possible bias.

Lab effects analysis was added to Material and methods and Supplementary, Lines 168-178. Temporal effect is mentioned in the text, Lines 318-320.

4) This study includes two types of SNP data: MH genotypes and putative neutral genotypes. I found it hard to follow in the Results which type (or both) of data were being reported.  I suggest adding text to always make it clear which SNPs are being referred to.

Thank you for this notice. All necessary text I added to the Material and methods and Results, Lines 204-205, 215-216 etc.

 5) The first section of the Discussion deals with the unexpected low LD between the two MH SNPs in many populations. While this is curious, it was not the aim of this work and it certainly is not the main outcome.  I suggest reducing this Discussion and moving it lower in the Discussion.

Done, the text has been shortened and moved to the Results.

 6) I like the Discussion of the observed clines in MH diversity, and both temperature and pathogen community could contribute to that pattern.  However, the author does not present actual pathogen data, hence I suggest this discussion be a bit more cautious (although the author does acknowledge that other factors may play a role). In any case, the author should consider putting this discussion text close to the beginning of the Discussion.

Corrected according to comments

 7) The first line of the Conclusion indicates that these data support balancing selection as a key factor in MH genotype/allele evolution in sockeye salmon. I suggest this be added as a key goal of the paper in the Introduction. As it stands the paper is framed as a report of observations of genotype/allele frequency distributions; however, the analysis is much more than that!

Thank you for recommendation, I followed it.

 8) While the paper is generally well-written, I suggest the grammar and word use be carefully checked – in places it is hard to follow and in others I think the text means something different than the author intended.

Thank you for understanding! I tried to do my best to improve it.

Specific comments

  • The final sentence of the abstract is hard to follow and needs to be clarified.

I tried to improve it.

  • The author rarefied the data to N >9 based on the smallest samples size, but in Table 1, the smallest reported is 15. Furthermore, most sample sizes were much larger than that, so it seems overly cautious to rarefy to N=10.

It was just a citation from a manual, I made corrections.

  • This study includes 29 putatively neutral SNPs used as the neutral control data; however, it seems some of those SNPs may not be entirely neutral. I think text needs to be added describing the neutral SNPs and an assessment of whether they are evolving by drift only added.

The description of methods of SNP selection was added in Material an Methods, Lines 188-196. Moreover the table of used loci was added to Supplementary.

  • Figure 4B: the concave nature of the relationship between allele frequency and latitude seems to be driven by a single population, perhaps the author should be cautious in interpreting this pattern.

I absolutely agree with the remark, I tried to avoid conclusions based on it.

  • Line 406-407: I don’t recall seeing the results of the Mantel test for the neutral SNP data? It may have missed it as I found it hard to know which data type was being reported.

Detailed information has been added, Lines 356-359.

  • Figure 4: perhaps a typo? “hitmap” = heatmap
  • Line 503: I suggest sticking with the generally accepted significance report (not P>0.99)

All typos were corrected.

  • Lines 535-536: “colonization by small number and less polymorphic…” why would the colonizing individual be less polymorphic? Generally the range edge individuals are more diverse, at least at MHC loci.

Corrected according to comments

I hope that I have made all the recommended edits to the text. I appreciate the special attention to the manuscript and the time spent on such valuable comments.

With my great respect and gratitude,

Anastasia Khrustaleva

Round 2

Reviewer 1 Report

Thank you for considering my comments. I think the revised manuscript is sufficient to solve my queries.